# Alterations in the inflammatory markers of the Tumor Necrosis Factor system in overweight and obese children and adolescents

Heloísa Marcelina da Cunha Palhares[1]☉, Adriana Paula da Silva[1]☉,
Janaíne Machado Tomé[1]☉, Marcos Vinícius da Silva[2]‡, Virmondes Rodrigues Júnior[2]‡,
Flávia Alves Ribeiro[1]☉, Marília Matos Oliveira[1]☉, Elvi Cristina Rojas Fonseca[1]☉,
Ianessa Arantes Valle[1]☉, Maria de Fátima Borges[1]☉*

1 Department of Endocrinology and Metabolism, Federal University of Triângulo Mineiro/ Clinical Hospital, Uberaba, Minas Gerais, Brazil, 2 Department of Microbiology, Immunology and Parasitology/ Federal University of Triângulo Mineiro/ Clinical Hospital, Uberaba, Minas Gerais, Brazil

☉ These authors contributed equally to this work.
‡ These authors also contributed equally to this work.
* borgmf@uol.com.br

## Abstract

### Objective

This study analyzed the association between cardiometabolic risk markers and the tumor necrosis factor system in overweight and obese children and adolescents.

### Methods

This cross-sectional study included 201 overweight (n = 65), obese (n = 96), and eutrophic (n = 40) children and adolescents aged 5 to 19 years. Clinical markers (body mass index, percentage of body fat, waist circumference, systolic and diastolic blood pressures) and laboratory parameters (glucose, insulin, total cholesterol and fractions, triglycerides, homeostasis assessment of insulin resistance index [HOMA-IR], leptin, tumor necrosis fator-α [TNF-α], soluble TNF receptors [sTNFR1 and sTNFR2], soluble Tumor necrosis factor-Related Apoptosis-Inducing Ligand [sTRAIL]) were evaluated.

### Results

Serum TNF-α levels did not differ significantly between the participant groups, while the serum concentrations of sTNFR1 were higher in the obesity group, compared with those in the eutrophic and overweight groups. Regarding sTNFR2, there was no significant difference between the three study groups. Serum sTRAIL concentrations were higher in the eutrophic group compared with those in the overweight and obesity groups. We observed a positive correlation between sTNFR1 and body mass index, waist circumference, triglycerides, glucose and leptin levels. There was also

**Data availability statement:** All relevant data are within the paper and its Supporting Information files.

**Funding:** The author(s) received no specific funding for this work.

**Competing interests:** The authors have no conflicts of interest to declare.

a negative correlation between sTRAIL and body mass index, waist circumference, LDL cholesterol, glucose and HOMA-IR levels.

## Conclusions

Inflammatory changes involving the TNF system (sTNFR1, sTRAIL) that correlate with obesity are present since childhood, indicating the need for early intervention in order to avoid cardiometabolic complications in adulthood.

---

## Introduction

The prevalence of childhood obesity has been steadily increasing in recent decades, both in developed and developing countries, and is a major global public health problem [1]. Adipose tissue is a metabolically active organ and produces several bioactive substances involved in the metabolic, endocrine, and immunological processes [2]. Excess body fat leads to metabolic changes in adipose tissue, induction of insulin resistance in insulin responsive cells (adipocytes, hepatocytes, myocytes, and β-cells) and endothelial dysfunction, through pro-inflammatory and prothrombotic effects resulting from the action of the inflammatory cytokines and adipokines [3, 4].

Tumor necrosis factor-α (TNF-α), a major proinflammatory mediator, acts directly on adipocytes that regulate fat accumulation, and on insulin-dependent processes such as glycemic homeostasis and lipid metabolism [5]. Studies in adults and obese children have revealed an increase in circulating TNF-α levels, as well as within adipose tissue [6, 7]. Conversely, studies in adults and children found that serum TNF-α levels did not change with weight gain, suggesting that circulating TNF-α levels do not reflect concentrations within adipose tissue [8–11].

In humans, two cell surface receptors for TNF-α, namely type 1 (TNFR1) and type 2 (TNFR2) have been described that undergo proteolytic cleavage of their extracellular parts upon ligand binding and give rise to soluble forms of the receptors, namely sTNFR1 and sTNFR2 [12]. It is suggested that these soluble forms can compete with cell surface receptors and inhibit TNF-α activity, or act to stabilize its structure, preserving its activity and potentiating some of its effects through a gradual process of binding and release of this cytokine [13]. Until now, studies evaluating serum concentrations of sTNFRs in obese children and adolescents have led to discordant results [9, 14, 15]. Pamir et al. (2009) reported a physiological role for sTNFRs by limiting body weight and adiposity through increased metabolic rate, fatty acid oxidation, and the infiltration of macrophages into adipose tissue with an inflammatory phenotype; however, this role was observed to be inoperable in the development of insulin resistance induced by obesity [16].

TNF-Related Apoptosis-Inducing Ligand (TRAIL) is a member of the TNF superfamily and is expressed as a transmembrane protein of various cell types or as a soluble protein (sTRAIL) [17]. Its primary biological activities include the induction of apoptosis, participating in the homeostasis of the immune system, and being a promising candidate for the treatment of malignant diseases, since it was able to induce selective apoptosis in cancer cells without reaching normal cells [17, 18]. Further, TRAIL

regulates homeostasis of adipose tissues and promotes proliferation of preadipocytes [19]. Some studies have suggested a relationship between TRAIL concentrations and adiposity, insulin resistance, and metabolic indices. In healthy adult persons older than 50 years, serum TRAIL concentrations were observed to be associated with body composition and serum lipids, correlating significantly with total fat in men and with total cholesterol and LDL-cholesterol in women [20, 21]. In addition, TRAIL was positively correlated with fat mass and waist circumference (WC) in individuals without apparent cardiovascular and metabolic disease [22] and correlated with body mass, insulin resistance and triglycerides in individuals with type 2 diabetes [23]. However, there are no studies focusing on TRAIL in normal or overweight children and adolescents.

The increase in the prevalence of childhood obesity is a concern and the mechanisms involving the pathophysiology of obesity are complex, with low-grade chronic inflammation, abnormal cytokine and adipokine production, and insulin resistance playing roles, which are responsible for the metabolic repercussions evident during childhood. Thus, we aimed to analyze the association between clinical, laboratory, and inflammatory biomarkers of the TNF system and body weight in overweight and obese children and adolescents.

## Materials and methods

A previous study was carried out in the city of Uberaba, Minas Gerais, Brazil from March 2012 to September 2013, where 1,125 school children and adolescents (5 to 19 years of age) were evaluated, where 916 were from public schools and 209 were from private schools; 364 participants had inadequate body weight, while 195 were overweight (17%) and 169 were obese (15%) [24]. The project was approved by the Committee of Ethics in Research with Human Beings of UFTM (protocol no. 2479).

The study participants were invited to attend the Triangulo Mineiro Federal University (UFTM) Pediatric Endocrinology Outpatient Clinic, to carry out medical consultations, laboratory tests, and receive guidance on the treatment of excess weight.

However, only 172 children (5 to 9 years) and adolescents (10 to 19 years old) attended the consultation. All 172 participants were examined by the same doctor, submitted to the collection of pertinent laboratory tests, and nutritional guidance was provided by a nutritionist along with drug treatment when necessary. During the consultation, they were invited to participate in the research project conducted from February 2013 to July 2014. Ten participants refused to participate, and we excluded one case due to the diagnosis of *Osteogenesis Imperfecta*. Thus, the final sample consisted of 161 overweight and obese children and adolescents. The eutrophic group consisted of 40 children and adolescents. A cross-sectional study was carried out.

In order to analyze the clinical markers, a nutritional assessment (body weight, height, body mass index [BMI] calculation, skinfolds and body fat percentage [BFP], WC, systolic and diastolic blood pressures [SBP and DBP], respectively) and pubertal staging were performed using standard procedures [25].

In order to classify the nutritional status of children and adolescents, a BMI-z score was used, according to the criteria proposed by the World Health Organization (WHO) [25], calculated using WHO-Anthro Plus 2007 (Geneva-Switzerland). The individuals were categorized into eutrophic (-2 ≤ BMI z score <+ 1), overweight (+1 ≤ BMI z score <+ 2) and obese (BMI z score ≥ + 2).

In eutrophic, overweight and obese children and adolescents, blood samples were collected by peripheral venous puncture after 10 to 12 hours of fasting for the measurement of glycemia, insulin, total cholesterol, HDL-cholesterol, LDL-cholesterol (LDL-c), and triglycerides (TG). Leptin and inflammatory cytokines (sTRAIL, TNF-α) and TNF-α receptors (sTNFR1, sTNFR2) were quantified in these groups.

The serum concentrations of total cholesterol, HDL-c, LDC-c and triglycerides were measured using an enzymatic colorimetric method and fasting glycemia by the enzymatic method with Hexokinase. All samples were processed in the COBAS 6000-module C501 (Roche Diagnóstica - São Paulo - Brazil). Insulin was measured by electrochemiluminescence (COBAS 6000-module C601 Roche Diagnóstica - São Paulo - Brazil). Leptin was assayed using enzyme linked immunosorbent assay (ELISA) in duplicate with a commercial EMD kit supplied by Millipore Corporation (Billerica, MA, USA).

The HOMA-IR index was obtained by calculating the fasting plasma insulin product (µU/mL) and fasting glycemia (mmol/L) divided by 22.5. The cytokines sTRAIL and TNF-α and their soluble receptors (sTNFR1 and sTNFR2) present in the plasma of overweight and eutrophic patients were quantified by the ELISA method (ENSPIRE - PERKIN ELMER - EUA). Concentrations of the cytokines and TNF receptors were determined from linear regression with the absorbances obtained in the recombinant cytokine curve and expressed in pg/mL. The minimum sensitivity threshold of the method ranged from 4 to 20 pg/mL.

For the statistical analysis, the Kolmogorov Smirnov test with Liliefors correction and the Levene test were used. Descriptive analysis was performed from absolute and percentage frequencies and descriptive measures of centrality and dispersion.

In the comparisons between three independent groups, an analysis of variance (ANOVA) was used followed by the multiple comparison test, namely ANOVA-F followed by the Tukey test or the Kruskall-Wallis test followed by the Dunn's test. In the comparisons between two independent groups, the Student's t-test or the Mann-Whitney test was used.

Correlations between two numerical variables were analyzed from the Pearson or Spearman linear correlation coefficients.

The level of significance for performing all inferential procedures was 5%. The STATISTICA program, Statsoft, version 10 was used to perform the statistical procedures.

## Results

A nutritional assessment of overweight children and adolescents showed that 40.4% (65/161) were overweight and 59.6% (96/161) were obese. The clinical and laboratory profiles of the eutrophic, overweight, and obesity groups are presented in Table 1.

**Table 1. Clinical and laboratory characterization of groups of eutrophic, overweight. obese children, and adolescents screened from public schools.**

| VARIABLE | GROUPS | | |
|---|---|---|---|
| | **Eutrophic** | **Overweight** | **Obesity** |
| | **(n = 40)** | **(n = 65)** | **(n = 96)** |
| Age (years) | 10.2 ± 2.9 | 12.6 ± 3.4 | 10.9 ± 2.5 |
| Sex (M/F) | 9/31 | 19/46 | 39/57 |
| Pubertal stage (pre-pubertal/pubertal) | 20/20 | 17/48 | 47/49 |
| BMI (kg/m$^2$) | 17.0 ± 1.9 | 23.0 ± 3.1 | 27.1 ± 3.7 |
| BMI z-score | 0.06 ± 0.7 | 1.6 (1.0 − 1.9) | 2.57 (2.0 − 4.9) |
| Waist circumference (cm) | 59.4 ± 5.5 | 73.4 ± 8.4 | 84.0 ± 10.7 |
| Percent body fat | | 34.5 ± 9.0 | 43.1 ± 8.4 |
| Systolic blood pressure (mmHg) | 102.7 ± 9.61 | 106.4 ± 9.7 | 108.7 ± 10.0 |
| Diastolic Blood Pressure (mmHg) | 64.2 ± 8.0 | 68.6 ± 7.6 | 70.3 ± 7.9 |
| Total cholesterol (mg/dL) | 126.8 ± 28.1 | 159.0 ± 33.5 | 169.9 ± 32.5 |
| HDL- cholesterol (mg/dL) | 45.8 ± 8.2 | 46.7 ± 10.8 | 44.6 ± 11.2 |
| LDL- cholesterol mg/dL | 67.9 ± 25.6 | 94.6 ± 29.6 | 105.5 ± 30.3 |
| Triglycerides (mg/dL) | 68.6 ± 31.2 | 81.0 (38.0 − 196.0) | 91.0 (31.0 − 445.0) |
| Glucose (mg/dL) | 76.9 ± 12.9 | 85.4 ± 12.1 | 87.6 ± 11.5 |
| Insulin (µIU/mL) | 8.4 ± 5.7 | 11.6 (3.5 − 32.8) | 14.5 (1.4 − 117.2) |
| HOMA-IR | 1.7 ± 1.3 | 2.3 (0.7 − 7.1) | 3.2 (0.3 − 28.7) |

Source: the author

The profiles of leptin, TNF-α, TNF receptors, and sTRAIL for the eutrophic, overweight and obesity groups are presented in Table 2.

An analysis of serum leptin concentrations showed a statistically significant difference between the three groups (p <0.0001), being higher in the obesity group in relation to the overweight and eutrophic groups, and also higher in the overweight compared to the eutrophic group (Table 2; Fig 1).

Serum concentrations of TNF-α were below the minimum limit detectable by the method used in 95% (38/40) of the eutrophic individuals, 93.8% (61/65) of overweight individuals, and 95.8% (92/95) of the individuals with obesity. In this case too, there was no statistically significant difference between the groups (p = 1.0; Table 2).

In contrast, serum concentrations of sTNFR1and sTNFR2 were detected in all samples analyzed. Regarding sTNFR1 serum concentrations, there was a statistically significant difference between the eutrophic and obesity groups (p = 0.004) and between the overweight and obesity groups (p = 0.006), the mean serum concentrations being higher in the obesity group in both analyses (Table 2). There was no statistically significant difference between the eutrophic and overweight groups (Fig 2). Regarding sTNFR2, there was no statistically significant difference between serum concentrations of the three groups (Table 2). An analysis of the concentrations by sex and pubertal staging did not show differences for both TNF-α receptors (p >0.05).

Serum quantification of sTRAIL levels was carried out in 60% (24/40) of the serum samples from the eutrophic group, 33.8% (22/65) of the overweight group, and 33.3% (32/96) of the obesity group. There was a statistically significant difference between the eutrophic group and the overweight group (p = 0.03) and between the eutrophic group and the obesity group (p = 0.008), with higher serum concentrations in the eutrophic group compared with the other groups in both analyses (Table 2). There was no statistically significant difference between the overweight and obesity groups (Fig 3).

**Table 2. Profile of leptin and cytokines of children and adolescents according to the eutrophic, overweight and obese groups screened from public schools.**

| VARIABLES | GROUPS | | | p§ | p† | pΦ |
|---|---|---|---|---|---|---|
| | Eutrophic (n=40) | Overweight (n=65) | Obesity (n=95) | | | |
| | Median (min-max) | Median (min-max) | Median (min-max) | | | |
| Age (years)[a] | 10.2 ± 2.9 10.2 (5.2 − 16) | 12.6 ± 3.4 12.2 (6.6 − 19.2) | 10.9 ± 2.5 10.7 (5.6 − 18.7) | 0.001 | 0.86 | 0.005 |
| Leptin (ng/mL)[a] | 3.8 ± 2.7 3.6 (0.5 − 10.7) | 15.2 ± 9.4 12.9 (1.3 − 38.9) | 24.1 ± 11.4 21.6 (3 − 55) | <0.0001 | <0.0001 | <0.0001 |
| TNF-α (pg/mL)[a] | 2.44 ± 15.43 0.0 (0.0 − 97.6) | 1.8 ± 8.6 0.0 (0.0 − 63.8) | 0.7 ± 4.9 0.0 (0.0 − 43.5) | 1.0 | 1.0 | 1.0 |
| sTNFR1 (pg/mL)[a] | 705.2 ± 341.2 586.5 (220 − 1753) | 778.5 ± 442.9 644.0 (263 − 2086) | 955.3 ± 451.3 862 (236 − 2464) | 1.0 | 0.004 | 0.006 |
| sTNFR2 (pg/mL)[a] | 1323.6 ± 221.8 1313.5(885 − 1967) | 1315.3 ± 379.7 1249 (489 − 2147) | 1348.9 ± 288.8 1324 (730 − 2110) | 1.0 | 1.0 | 0.7 |
| sTRAIL (pg/mL)[a] | 90.9 ± 235.3 33.5 (0.0 − 1397.2) | 32.7 ± 86.5 0.0 (0.0 − 585.8) | 46.1 ± 204.4 0.0 (0.0 − 1477.5) | 0.03 | 0.008 | 1.0 |

Source: the author.

[a]Kruskal-Wallis test: values expressed as medians (Vmin - Vmax);

§Eutrophic x overweight;

†Eutrophic x obesity;

ΦOverweight x Obesity

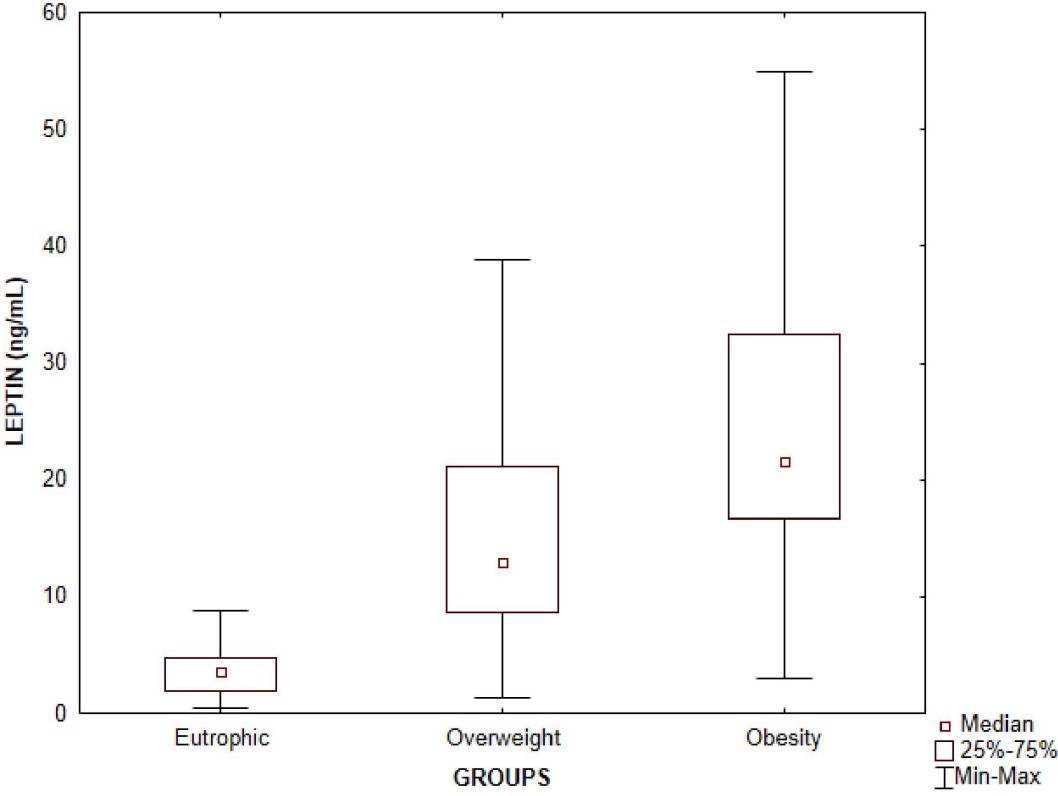

**Fig 1. Evaluation of serum leptin concentrations in serum samples of children and adolescents according to the eutrophic, overweight and obesity groups.** The symbols represent the medians, the bars represent the percentiles 25 to 75 and the vertical lines represent the minimum and maximum values. Significant differences between the eutrophic and overweight groups (p < 0.0001), eutrophic and obesity (p < 0.0001), overweight and obesity (p < 0.0001). Kruskal-Wallis test followed by Dunn's multiple comparison.

We carried out a simple linear correlation analysis of the cytokines (TNF-α, sTRAIL) and soluble TNF receptors (sTNFR1, sTNFR2) with age, clinical parameters (BMI, BMI z score, % body fat, WC, SBP, DBP), laboratory parameters (total cholesterol, LDL-c, HDL-c, TG, insulin, HOMA-IR) and leptin levels. The results of this analysis that exhibited statistical significance are shown in Table 3.

## Discussion

The complications associated with excessive weight gain are evident from childhood, and an increase in adipose tissue promotes changes in adipokines, such as elevated leptin levels. Among the metabolic changes triggered by obesity in adipose tissue is low-grade systemic inflammation, which is an important cause of insulin resistance that leads to changes in the profile of inflammatory biomarkers, such as those observed in sTNFR1 and sTRAIL levels in the present study. Several proinflammatory markers are elevated, while anti-inflammatory markers are usually reduced in obese children. An imbalance between these markers contributes to the undesirable metabolic effects observed in obesity [26].

In this context, high concentrations of TNF-α were observed in obese children and adolescents [7, 27 ,28]. Conversely, Hauner et al. found no differences in serum TNF-α concentrations among lean, obese, and obese diabetic adults and speculated that TNF-α overexpression within adipose tissue does not correlate with circulating TNF-α concentrations, indicating a primarily local role for this cytokine [8]. This finding was subsequently observed in several other studies on obese and lean prepubertal children [9–11, 29]. In the present study, we did not observe differences in circulating TNF-α

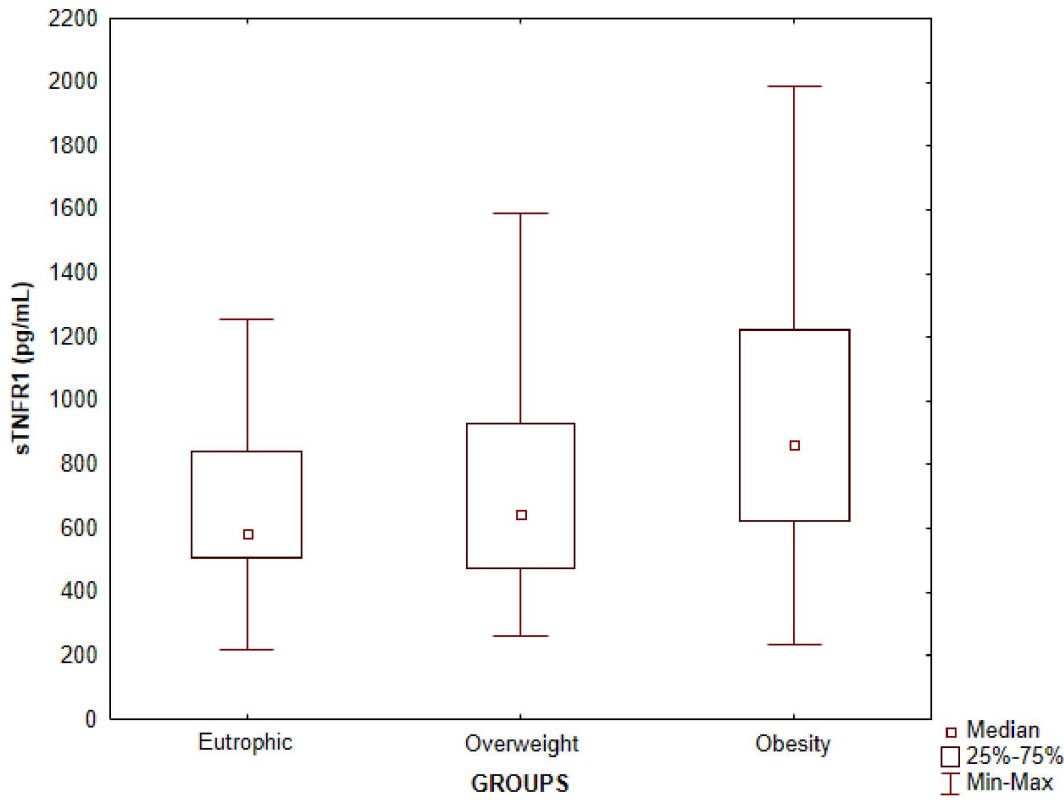

**Fig 2. Evaluation of serum concentrations of sTNFR1 in plasma samples of children and adolescents according to the eutrophic, overweight and obesity groups.** The symbols represent the medians, the bars represent the percentiles 25 to 75 and the vertical lines represent the minimum and maximum values. Significant differences between the eutrophic and obesity groups (p = 0.004), overweight and obesity (p = 0.006). Kruskal-Wallis test followed by Dunn's multiple comparison.

concentrations among the various participant groups, where almost all variations in levels were below the detection limit of the method; however, we did not rule out the possibility of local elevations within adipose tissue leading to autocrine and paracrine effects.

The plasma concentrations of soluble TNF-α receptors appear to reflect the degree of TNF activation [13]. In the present study, sTNFR1 and sTNFR2 analyses showed higher sTNFR1 serum concentrations in the obese group compared with those of the overweight and eutrophic groups, while sTNFR2 levels did not differ among the groups. This finding is in agreement with the work of Ouyang et al., who compared obese and lean prepubertal children (3–4 years) and found elevated sTNFR1 serum concentrations in obese children while sTNFR2 levels remained unchanged [14]. Importantly, we observed a significant correlation between sTNFR1 levels and waist circumference, and between sTNFR2 concentration and the percentage of body fat, suggesting the influence of visceral and total adiposity on these receptors.

Notably, other studies evaluating TNF-α receptors in obese adults, children and adolescents have shown discordant results. Some reports showed elevation of both sTNFR1 and sTNFR2 levels [ 5, 8], one report found an increase in sTNFR2 alone [30], while other studies showed no alteration in these receptors to be associated with weight gain [ 6, 15]. Some receptor studies showed an increase in sTNFR1 [31], or sTNFR2 [32]. This discrepancy in the findings of studies related to sTNFR1 and sTNFR2 is attributed to variations in the obesity phenotype, degree of obesity, sex, age, environmental and genetic factors [ 14, 33, 34].

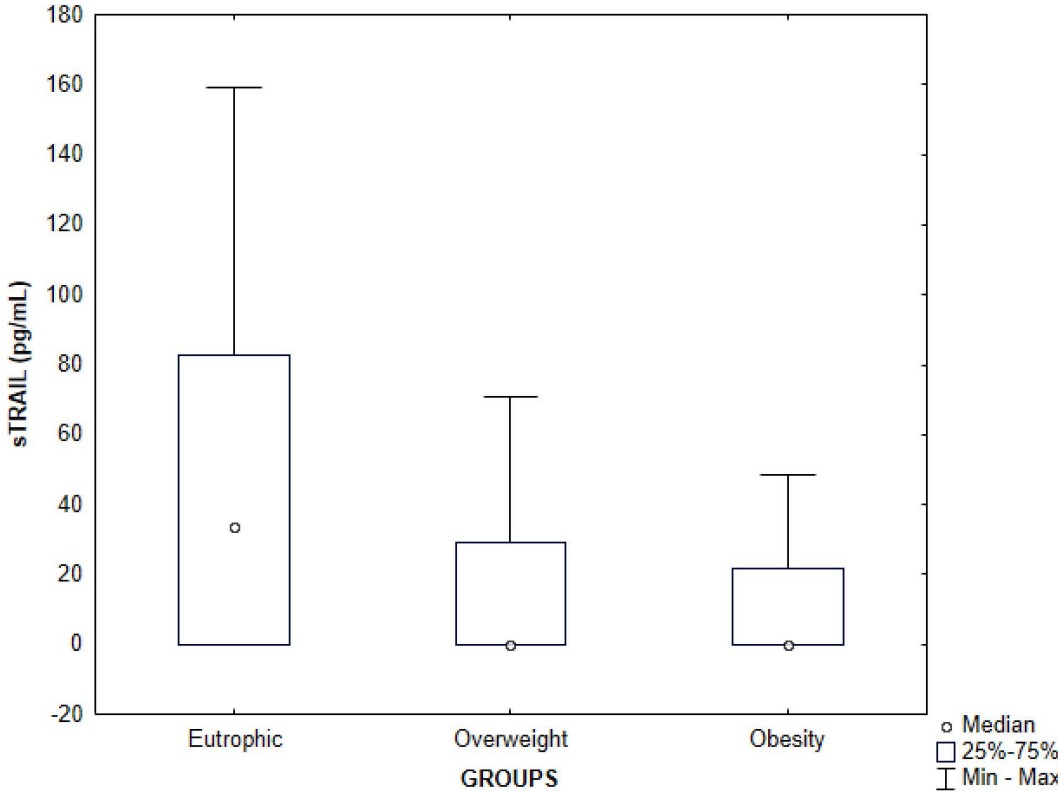

**Fig 3. Evaluation of serum sTRAIL concentrations in plasma samples of children and adolescents according to the eutrophic, overweight and obesity groups.** The symbols represent the medians, the bars represent the percentiles 25 to 75 and the vertical lines represent the minimum and maximum values. Significant differences between the eutrophic and overweight groups (p = 0.03), eutrophic and obesity (p = 0.008). Kruskal-Wallis test followed by Dunn's multiple comparison.

**Table 3. Simple linear correlation (r) between soluble TNF receptors (sTNFR1, sTNFR2), Tumor Necrosis Factor-Related Apoptosis-Inducing Ligand (sTRAIL) and clinical, laboratory parameters including leptin, in the total group of eutrophic, overweight and obese children and adolescents screened from public schools.**

| VARIABLE# | VARIABLE | | | | | |
|---|---|---|---|---|---|---|
| | sTNFR1 | | sTNFR2 | | sTRAIL | |
| | r | p | r | p | r | p |
| BMI | 0.22 | 0.001 | 0.03 | 0.63 | -0.16 | 0.02 |
| BMI z-score | 0.33 | <0.0001 | 0.05 | 0.51 | -0.19 | 0.006 |
| Waist circumference | 0.25 | 0.0005 | 0.11 | 0.12 | -0.15 | 0.03 |
| Body fat percentage | 0.11 | 0.16 | 0.19 | 0.01 | 0.10 | 0.19 |
| LDL- cholesterol | 0.08 | 0.27 | -0.05 | 0.44 | -0.14 | 0.04 |
| Triglycerides | 0.17 | 0.02 | 0.14 | 0.05 | 0.04 | 0.54 |
| Glucose | 0.24 | 0.0007 | 0.008 | 0.90 | -0.16 | 0.03 |
| HOMA-IR | 0.09 | 0.22 | 0.04 | 0.58 | -0.15 | 0.03 |
| Leptin | 0.21 | 0.004 | 0.07 | 0.32 | -0.08 | 0.28 |

Source: the author

#Spearman Linear Correlation Coefficient

Evaluating the relationship between TNF- α receptors and adipokines we observed that sTNFR1 is positively correlated with leptin, in agreement with other authors, which suggested direct modulation of leptin secretion by sTNFR1 [31, 35]. Furthermore, other studies have shown that leptin positively regulated TNF-α expression in obese individuals and in human mononuclear cells [36, 37]. Previously, Chu et al. showed that sTNFR1 was positively associated with leptin only in females and suggested that sTNFR1 could play a significant role in leptin expression in girls only [31]. Although we did not observe any difference between groups regarding sTNFR2, the correlation of sTNFR2 with percentage of body fat suggests that this receptor changes with the progress of total adiposity.

Since TNF-α effects predominantly work via cell surface TNFR1 and high sTNFR1 concentrations in the obese group correlate with leptin levels, we speculate that sTNFR1 may be indirectly reflecting the local increase of TNF-α within adipose tissue. Thus, we hypothesize that weight gain may elevate TNF-α within adipose tissue and its binding to TNFR1 on the cell surface may generate a systemic increase in sTNFR1 levels, secondary to the local action of TNF-α.

In the children and adolescents studied here, we observed higher serum sTRAIL concentrations in the eutrophic group compared with the overweight and obese groups; however, this cohort did not display the metabolic repercussions associated with sTRAIL elevations. We observed a weakly significant negative correlation with BMI and WC in the group as a whole. We speculate that if in the early stages of weight gain in childhood and adolescence, sTRAIL levels decrease like some anti-inflammatory cytokines, the ensuing weight gain and resulting cardiovascular disease may lead to a reversal of this profile and increased sTRAIL levels as a protective response to the comorbidities that accompany obesity.

The positive association of TRAIL with deleterious cardiometabolic abnormalities, such as adiposity, and the simultaneous inverse association with the risk of cardiovascular disease mortality [38, 39], raises questions and hypotheses. It has been speculated that elevated sTRAIL levels may protect overweight and obese individuals (already affected by cardiovascular disease) from the occurrence of future cardiovascular events through an unspecified adaptive mechanism [17]. This hypothesis is reinforced by the direct relationship of sTRAIL with adiposity and cardiovascular risk markers such as BMI, WC, fat mass, total cholesterol, LDL-cholesterol, triglycerides and an inverse correlation with HDL-cholesterol [21]. A study in patients with cardiovascular disease showed that high concentrations of TRAIL have a protective effect on the risk of future cardiovascular events and mortality [38, 39].

The cellular source of both TRAIL and sTRAIL as well as sTRAIL secretion mechanisms are not fully understood and it is unclear whether the observed direct correlation of TRAIL concentration with obesity reflects increased synthesis and/or increased adipocyte release or is a consequence of the activation of other biological pathways by adipose tissue [21]. New research in the pediatric age group, as well as longitudinal studies, may clarify and point out possible roles played by TRAIL during the evolution of changes in body composition and their metabolic repercussions, as well as the role of sTRAIL as a regulator of adipokines.

Inflammatory changes in the TNF system related to obesity that have been identified in overweight and obese children and adolescents reinforce the need for intervention with educational measures focused on weight loss. In addition, a better understanding of the mechanisms involving inflammation, adipose tissue and their cardiometabolic repercussions in the early stages of weight gain may elucidate possible therapeutic targets in the pediatric population and especially in adults. Diagnosing and treating childhood obesity and its comorbidities are challenges that should be part of pediatric public health programs in order to avoid cardiovascular and metabolic repercussions in adult life.

Our study has some limitations, as it is a cross-sectional assessment and temporal relationships cannot be definitively established. However, there is strong evidence that excess adipose tissue is a causal factor in the appearance of cardiometabolic risk factors since childhood.

## Supporting information

**S1 Data. Database with statistics control**.
(XLS)

## Author contributions

**Conceptualization:** Heloísa Marcelina da Cunha Palhares, Maria de Fátima Borges.

**Data curation:** Adriana Paula da Silva, Janaíne Machado Tomé, Marcos Vinícius da Silva, Virmondes Rodrigues Júnior, Flávia Alves Ribeiro, Marília Matos Oliveira, Elvi Cristina Rojas Fonseca, Ianessa Arantes Valle.

**Formal analysis:** Heloísa Marcelina da Cunha Palhares, Adriana Paula da Silva, Janaíne Machado Tomé, Maria de Fátima Borges.

**Investigation:** Heloísa Marcelina da Cunha Palhares, Marcos Vinícius da Silva, Virmondes Rodrigues Júnior, Flávia Alves Ribeiro, Maria de Fátima Borges.

**Methodology:** Heloísa Marcelina da Cunha Palhares, Adriana Paula da Silva, Janaíne Machado Tomé, Maria de Fátima Borges.

**Project administration:** Heloísa Marcelina da Cunha Palhares, Adriana Paula da Silva, Maria de Fátima Borges.

**Resources:** Maria de Fátima Borges.

**Software:** Maria de Fátima Borges.

**Supervision:** Heloísa Marcelina da Cunha Palhares, Maria de Fátima Borges.

**Validation:** Maria de Fátima Borges.

**Visualization:** Maria de Fátima Borges.

**Writing – original draft:** Heloísa Marcelina da Cunha Palhares.

**Writing – review & editing:** Heloísa Marcelina da Cunha Palhares.

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
