## [Decision Letter · Decision Letter 0]

23 Sep 2024

PONE-D-24-30787Alterations in the inflammatory markers of the Tumor Necrosis Factor system in overweight and obese children and adolescents.PLOS ONE

Dear Dr. RIBEIRO,

Thank you for submitting your manuscript to PLOS ONE. After careful consideration, we feel that it has merit but does not fully meet PLOS ONE’s publication criteria as it currently stands. Therefore, we invite you to submit a revised version of the manuscript that addresses the points raised during the review process.

Please attend to the comments from reviewer 2 as suggested.

We look forward to receiving your revised manuscript.

Kind regards,

Sepiso K. Masenga, PhD

Academic Editor

PLOS ONE

Journal Requirements:

“The authors have no conflicts of interest to declare.”

3. We note that your Data Availability Statement is currently as follows: [If the data are all contained within the manuscript and/or Supporting Information files, enter the following: All relevant data are within the manuscript and its Supporting Information files.]

If your submission does not contain these data, please either upload them as Supporting Information files or deposit them to a stable, public repository and provide us with the relevant URLs, DOIs, or accession numbers. For a list of recommended repositories, please see https://journals.plos.org/plosone/s/recommended-repositories .

Reviewers' comments:

Reviewer's Responses to Questions

**Comments to the Author**

1. Is the manuscript technically sound, and do the data support the conclusions?

Reviewer #1: Yes

Reviewer #2: Partly

2. Has the statistical analysis been performed appropriately and rigorously? 

Reviewer #1: Yes

Reviewer #2: N/A

3. Have the authors made all data underlying the findings in their manuscript fully available?

Reviewer #1: Yes

Reviewer #2: Yes

4. Is the manuscript presented in an intelligible fashion and written in standard English?

Reviewer #1: Yes

Reviewer #2: No

5. Review Comments to the Author

Reviewer #1: Title: Alterations in the Inflammatory Markers of the Tumor Necrosis Factor system in overweight and obese children and adolescents

Summary

This study by Palhares et. al investigates the relationship between obesity, inflammation, and metabolic markers in children, predominantly focusing on TNF-α and soluble TRAIL (sTRAIL). Though the study was unable to detect TNF-α in most samples, it reveals significant differences in metabolic markers between obese and non-obese children. Study findings suggest that inflammation plays a role in childhood obesity, though the precise mechanisms remain unclear. The present study highlights the need for more studies to determine sTRAIL’s role in obesity-related complications.

Comment

The article is well-written and presents a clear and focused investigation into the relationship between obesity, inflammation, and metabolic markers in children and adolescents. The authors have effectively explained the study's objectives, methods, and findings on obesity-related inflammation. Despite limitations, the study addresses important gaps in pediatric obesity research and suggests avenues for future exploration, particularly regarding inflammatory markers. The language is accessible, and the structure is logical, making the article a valuable contribution to the field.

Based on its scientific merit and clarity, the article deserves to be considered for publication.

Reviewer #2: Manuscript title: Alterations in the inflammatory markers of the Tumor Necrosis Factor system in overweight and obese children and adolescents.

Summary statement of the article

The article titled "Alterations in the Inflammatory Markers of the Tumor Necrosis Factor system in overweight and obese children and adolescents." This study explores the relationship between cardio-metabolic risk markers and the Tumor Necrosis Factor system in a diverse group of children and adolescents. In the study, authors analyzed various clinical and laboratory parameters to better understand the impact of overweight and obesity on inflammatory markers. The research aims to provide insights into the inflammatory responses linked to obesity in the pediatric population, contributing to a better understanding of the health implications of childhood obesity. The manuscript title and in itself is a scientific marvel to look at. However, some things can be looked at to improve the article.

Specific areas of improvement

Major comments:

1. Under materials and methods, lines 120-139 are a bit confusing with the grammar. Follow the heading and revise accordingly the entire section. I must the criteria of participant selection is confusing and given slim chances that it can replicated.

2. From Table 1, under the results section you have indicated 2 groups of interest contrary to the concept of 3 groups. Please indicate the characteristics of the eutrophic group as well.

3. Include the subheadings where necessary, it is not easy to tell which paragraph has clinical implications, conclusion/summary including the limitations of the study which could be expanded.

4. Results have not been clearly communicated, the discussion has the comparison factor to other relevant literature which the majority has been cited but the logic flow is a bit missing.

Minor comments:

1. Lines 184-188, “The project was approved by the Committee of Ethics in Research with Human Beings of UFTM (protocol no. 2479). Children, adolescents and parents and/or parents were clarified about the research project and those who agreed to participate of the study project signed the term of free and informed consent and the term of assent.” Move text to somewhere under statements of ethics and revise the sentence.

2. Spelling error I guess of “Eutrofic” in Figure 1. Please clarify.

3. Overall the grammar in the article is not so clear to easily understand, if authors can logically simplify the text and also there are some short paragraphs which can be merged into sizeable pieces, that would be great and could improve the clarity of the article.

4. Cite each statement that has been made in the article, as this covers authors on plagiarism. E.g. lines 67-71, 83-85. Please countercheck for others.

6. PLOS authors have the option to publish the peer review history of their article (what does this mean? ). If published, this will include your full peer review and any attached files.

**Do you want your identity to be public for this peer review?** For information about this choice, including consent withdrawal, please see our Privacy Policy .

Reviewer #1: No

Reviewer #2: No

---

## [Author Response · Author response to Decision Letter 1]

2 Nov 2024

Response to reviewers:

Manuscript title: Alterations in the inflammatory markers of the Tumor Necrosis Factor system in overweight and obese children and adolescents.

Larger comments:

Major comments:

1. Under materials and methods, lines 120-139 are a bit confusing with the grammar. Follow the heading and revise accordingly the entire section. I must the criteria of participant selection is confusing and given slim chances that it can replicated:

Response: In materials and methods, grammar was revised and the article was sent to Wiley Editing Services, as you can see in the certificate that we attached in the supplementary materials. The changes made are marked in the revised text.

Lines 129-139 were written as follows:

From February 2013 to July 2014, 172 children (5 to 9 years) and adolescents (10 to 19 years old) who were overweight and obese attended the consultation. All children and adolescents who consulted at the UFTM outpatient clinic were examined by the same doctor, submitted to the collection of pertinent laboratory tests, nutritional guidance provided by a nutritionist and drug treatment when necessary. During the consultation, they were invited to participate in the research project, with refusal by the same or responsible in 10 cases and exclusion from the study in 1 case due to the diagnosis of Osteogenesis Imperfecta. Thus, the final sample consisted of 161 overweight and obese children and adolescents. The eutrophic group consisted of 40 children and adolescents. A cross-sectional study was carried out.

Now the lines are written as follows:

A previous study was carried out in the city of Uberaba, Minas Gerais, Brazil from March 2012 to September 2013, where 1,125 school children and adolescents (5 to 19 years of age) were evaluated, where 916 were from public schools and 209 were from private schools; 364 participants had inadequate body weight, while 195 were overweight (17%) and 169 were obese (15%) [24].

The study participants were invited to attend the Triangulo Mineiro Federal University (UFTM) Pediatric Endocrinology Outpatient Clinic, to carry out medical consultations, laboratory tests, and receive guidance on the treatment of excess weight.

However, only 172 children (5 to 9 years) and adolescents (10 to 19 years old) attended the consultation. All 172 participants were examined by the same doctor, submitted to the collection of pertinent laboratory tests, and nutritional guidance was provided by a nutritionist along with drug treatment when necessary. During the consultation, they were invited to participate in the research project conducted from February 2013 to July 2014. Ten participants refused to participate, and we excluded one case due to the diagnosis of Osteogenesis Imperfecta. Thus, the final sample consisted of 161 overweight and obese children and adolescents. The eutrophic group consisted of 40 children and adolescents. A cross-sectional study was carried out.

2. From Table 1, under the results section you have indicated 2 groups of interest contrary to the concept of 3 groups. Please indicate the characteristics of the eutrophic group as well:

Response: Table 1, results section, was revised and we added the group of eutrophic patients as requested.

3. Include the subheadings where necessary, it is not easy to tell which paragraph has clinical implications, conclusion/summary including the limitations of the study which could be expanded:

Response: We reviewed the grammar and translation of the study to better understanding of all sections of the article. We saw no need to include subtitles.

4. Results have not been clearly communicated, the discussion has the comparison factor to other relevant literature which the majority has been cited but the logic flow is a bit missing:

Response: The discussion flow has been revised and we have added pertinent relevant literature.

Minor comments:

1. Lines 184-188, “The project was approved by the Committee of Ethics in Research with Human Beings of UFTM (protocol no. 2479). Children, adolescents and parents and/or parents were clarified about the research project and those who agreed to participate of the study project signed the term of free and informed consent and the term of assent.” Move text to somewhere under statements of ethics and revise the sentence.

Response: The text was moved to a place about ethics statement. However, one of the guidelines received in this email was that the ethics statement should only appear in the materials and methods section. Therefore, this guidance was a little contradictory because, in the guidance sent, it asks for this phrase to appear only in the methods section and in the reviewer's comments it asks for it to appear in an appropriate place.

2. Spelling error I guess of “Eutrofic” in Figure 1. Please clarify.

Response: Spelling error of the word eutrophic was corrected.

3. Overall the grammar in the article is not so clear to easily understand, if authors can logically simplify the text and also there are some short paragraphs which can be merged into sizeable pieces, that would be great and could improve the clarity of the article.

Response: Spelling and translation into English have been revised

4. Cite each statement that has been made in the article, as this covers authors on plagiarism. E.g. lines 67-71, 83-85. Please countercheck for others.

Response: A review of the article's references was carried out, the lines oriented for review are based on references 1; 2 e 12 that were added.

---

## [Decision Letter · Decision Letter 1]

9 Feb 2025

Alterações nos marcadores inflamatórios do sistema Fator de Necrose Tumoral em crianças e adolescentes com sobrepeso e obesidade.

PONE-D-24-30787R1

Dear Dr. RIBEIRO,

We’re pleased to inform you that your manuscript has been judged scientifically suitable for publication and will be formally accepted for publication once it meets all outstanding technical requirements.

Kind regards,

Sepiso K. Masenga, PhD

Academic Editor

PLOS ONE

Additional Editor Comments (optional):

Reviewers' comments:

Reviewer's Responses to Questions

**Comments to the Author**

1. If the authors have adequately addressed your comments raised in a previous round of review and you feel that this manuscript is now acceptable for publication, you may indicate that here to bypass the “Comments to the Author” section, enter your conflict of interest statement in the “Confidential to Editor” section, and submit your "Accept" recommendation.

Reviewer #1: All comments have been addressed

Reviewer #2: All comments have been addressed

2. Is the manuscript technically sound, and do the data support the conclusions?

Reviewer #1: (No Response)

Reviewer #2: Yes

3. Has the statistical analysis been performed appropriately and rigorously? 

Reviewer #1: (No Response)

Reviewer #2: Yes

4. Have the authors made all data underlying the findings in their manuscript fully available?

Reviewer #1: (No Response)

Reviewer #2: Yes

5. Is the manuscript presented in an intelligible fashion and written in standard English?

Reviewer #1: (No Response)

Reviewer #2: Yes

6. Review Comments to the Author

Reviewer #1: (No Response)

Reviewer #2: The authors have rigorously addressed the submitted review comments, resolving concerns to enhance the manuscript's clarity.

7. PLOS authors have the option to publish the peer review history of their article (what does this mean? ). If published, this will include your full peer review and any attached files.

**Do you want your identity to be public for this peer review?** For information about this choice, including consent withdrawal, please see our Privacy Policy .

Reviewer #1: No

Reviewer #2: No

---

## [Editor Report · Acceptance letter]

PONE-D-24-30787R1

PLOS ONE

Dear Dr. Ribeiro,

I'm pleased to inform you that your manuscript has been deemed suitable for publication in PLOS ONE. Congratulations! Your manuscript is now being handed over to our production team.

Kind regards,

on behalf of

Prof. Sepiso K. Masenga

Academic Editor

PLOS ONE